# OpenReview forum: "How does information access affect LLM monitors' ability to detect sabotage?"
_ICLR.cc/2026/Conference — Submitted to ICLR 2026_

### Official Review · Reviewer_iJFr · 2025-10-27

**Soundness:** 2
**Presentation:** 2
**Contribution:** 1
**Rating:** 2
**Confidence:** 3

**Summary:**

This paper studies the question of "how does information access affect LLM monitors' ability to detect sabotage?". They experimented across 3 environments and showed that monitors tend to exhibit what they called the "less-is-more" effect. Additionally, they proposed a new monitoring framework, "extract-and-evaluate," and claimed that this outperformed baselines in the BigCodeBench environment. Lastly, they conducted the cost-performance analysis.

**Strengths:**

- Ran 3 different environments
- The background is well motivated
- Conducted a diverse set of experiments

**Weaknesses:**

This paper claims to have 4 main contributions in Line 83, but I find 3 of them are either relatively weak or not original.
- Contribution 1 (less-is-more effect) is not a unique contribution; The result from Arnav et al. (arXiv:2505.23575) has already shown this.
- Contribution 2 (extract-and-evaluate monitoring) seems to be quite environment and monitor-dependent. In Fig. 3, based on the error bars, it seems to work noticeably well only for the gpt-4.1-mini monitor on the BigCodeBench and the MLE-sabotage. The other 5 monitor-environment combinations don’t show as substantial an improvement when considering the error bars.
- Contribution 3: Didn’t Arnav et al also show the experiment results with the reasoning and non-reasoning models as attackers?

**Questions:**

- For contribution 3, didn’t Arnav et al also show the experiment results with the reasoning and non-reasoning models as attackers?
- I'm happy to hear the authors' responses on the above weaknesses.

---

> ### Author Response · Authors · 2025-11-19
>
> Thank you for your review.
>
> We appreciate your critiques, but think that your concerns are either overstated or based on factual mistakes.
>
> ---
>
> ## Contribution 1
> It is true that Arnav et al. demonstrated one form of the less-is-more effect in one environment (action-only monitors can outperform CoT+action monitors in BigCodeBench-sabotage on non-subtle side tasks). We perform a much more systematic evaluation across many settings and monitor protocols, and our findings provide substantially stronger empirical evidence about the scope and nature of the less-is-more phenomenon. Specifically, we demonstrate that CoT-only and EaE monitors can also outperform CoT+action monitors and that some form of the less-is-more effect is true across all of BigCodeBench-sabotage, SHADE-Arena, and MLE-sabotage. We think it is important that people building black-box monitors take this (much stronger) evidence into account.
>
> ## Contribution 2
> In addition to the positive results you highlight for EaE for GPT-4.1-mini on BigCodeBench and MLE-Sabotage, note that EaE is significantly better than all other methods for Sonnet 3.7 in BigCodeBench-sabotage with a reasoning model attacker, and that it is never significantly worse than CoT+action monitors in our experiments. Furthermore, EaE performs especially well in the BigCodeBench-sabotage setting where side tasks are pursued by reasoning model attackers on subtle side tasks (see the updated Figure 4), sometimes more than doubling the catch rates of existing methods. Improving catch rates in this setting is especially important: we expect reasoning models to be adopted more widely in the future and to become more capable at executing misbehaviors in a subtle manner.
>
> We agree that the effectiveness of EaE is somewhat influenced by the environment and monitor. However, this seems true for many methods that are used in practice — it is unlikely that any single information-filtering method will be optimal in every use case. In practice, it may make sense for info-filtering to vary based on context, but carefully studying how to do context-dependent info-filtering is outside the scope of our project. Our work can serve as a valuable starting point for follow-up work in this direction, as it provides insights about strategies that are likely to work well with specific monitors and settings, such as EaE in long reasoning model trajectories with weaker monitor models.
>
> ## Contribution 3
> > Didn’t Arnav et al also show the experiment results with the reasoning and non-reasoning models as attackers?
>
> No! Arnav et al. used five models as attackers (see Figure 4 in Arnav et al.): Claude-3.7-Sonnet:Thinking, DeepSeek-R1, DeepSeek-R1-Distill-Llama-70B, DeepSeek-R1-Distill-Qwen-14B, and Grok-3-mini-beta. All of these are reasoning models. (In fact, our paper already says the following in Section 3: “As Arnav et al. (2025) only evaluated reasoning LLMs, we generated new attacker trajectories with two traditional LLMs: a traditional version of Claude-3.7-Sonnet and DeepSeekV3.”)
>
> ---
>
> As an additional point, we note that you gave us a low score for presentation. However, you did not comment on the presentation of our paper or point to any aspects which should be improved. Moreover, other reviewers commented very positively on our presentation. For example, reviewer uPer31 wrote: “The paper is very well-written and the presentation is overall well-done.”
>
> Overall, we think that the critiques you raise are not strong enough to warrant such a low score, and that you may be undervaluing some of the contributions of our paper. We hope the arguments we have provided here address your concerns and justify a higher score.

---

> > ### Comment · Reviewer_iJFr · 2025-11-19
> >
> > Thank you for the detailed responses.
> >
> > > We perform a much more systematic evaluation across many settings and monitor protocols, and our findings provide substantially stronger empirical evidence about the scope and nature of the less-is-more phenomenon.
> >
> > I acknowledge that the authors evaluated multiple setups using four different monitoring protocols, and that consistent results across these variations could strengthen the evidence for a “less-is-more” effect. However, upon examining Figure 3, I do not see strong empirical support for this claim either. In SHADE-Arena, the CoT + Action method outperforms both Action-only and CoT-only. In MLE-sabotage, the CoT + Action method also outperforms both Action-only and CoT-only for claude-3.7-sonnet. Taken together, these observations suggest that the results do not consistently support a less-is-more effect. Therefore, I think the claim that "CoT+action monitors often perform worse than monitors with less information across all environments" is not well supported by the data shown in the paper, nor do the authors explain why the above two settings do not have the less-is-more effect. There are some cases that present the less-is-more, but because it's not consistent across environments, I think the claim is a bit overstated.
> >
> > > We agree that the effectiveness of EaE is somewhat influenced by the environment and monitor. However, this seems true for many methods that are used in practice
> >
> > Thank you for the comment. My question is: if the effectiveness of EaE varies with the environment and the monitor (based on Fig 3), in what way should the readers view EaE as a useful contribution to the safety community? Since many existing methods also depend on these factors, how does EaE offer significant advantages beyond existing approaches?
> >
> > > No! Arnav et al. used five models as attackers (see Figure 4 in Arnav et al.): Claude-3.7-Sonnet:Thinking, DeepSeek-R1, DeepSeek-R1-Distill-Llama-70B, DeepSeek-R1-Distill-Qwen-14B, and Grok-3-mini-beta. All of these are reasoning models.
> >
> > Thank you for the explanation. I acknowledge that the difference here is that the authors also ran a non-reasoning LLM. And I agree that "CoT-only monitors perform significantly better for traditional than for reasoning LLMs" is a potentially useful finding. However, the authors draw this conclusion only from one setup (BigCodeBench-sabotage), and it is unclear to me whether this finding also generalizes to the other 3 setups in the paper.  I would hope to see more evidence beyond one environment. And drawing a general conclusion from one environment seems overreaching.
> >
> > > As an additional point, we note that you gave us a low score for presentation. However, you did not comment on the presentation of our paper or point to any aspects which should be improved.
> >
> > Thank you for this response. Upon examination, I agree that the presentation is decent, so I have raised the presentation score to 3.
> >
> > Unless the 3 main concerns mentioned above are addressed, I will respectfully maintain my original score for now.

---

> > > ### Author Response · Authors · 2025-11-19
> > >
> > > Thank you for the quick response!
> > >
> > > > However, upon examining Figure 3, I do not see strong empirical support for this claim either. In SHADE-Arena, the CoT + Action method outperforms both Action-only and CoT-only. In MLE-sabotage, the CoT + Action method also outperforms both Action-only and CoT-only for claude-3.7-sonnet.
> > >
> > > EaE is also an information filtering technique, so when EaE performs better than the CoT+action monitor, that’s an instance of the less-is-more effect! Out of the eight settings displayed in Figure 3, there is only one where CoT+action monitoring outperforms all three information filtering techniques (SHADE-Arena with a Claude-3.7-Sonnet monitor). When information filtering outperforms CoT+action monitoring, the difference can be dramatic (see Figure 4 in the main body and Figure 9 in Appendix B.1.1). This is never the case when CoT+action monitors outperform information filtering approaches, even when also considering the results with GPT-4.1 and Claude-3.5-Haiku monitors in Appendix B.1.1: CoT+action monitoring outperforms all information filtering methods only in two of the 16 settings displayed across Figure 3 and Figure 8 (Appendix B.1.1), and in neither of those two cases is the difference statistically significant. Furthermore, CoT+action monitoring never outperforms EaE by a statistically significant margin.
> > >
> > > For those reasons, we remain convinced that the claim "CoT+action monitors often perform worse than monitors with less information across all environments" is a good representation of our results: the results are mixed in SHADE-Arena, showing that CoT+action monitors don’t always perform worse than monitors with less information, but the effect can be observed in SHADE-Arena and occurs very consistently in the other environments. We intend to provide a more detailed account of why certain environments show stronger less-is-more effects than others before the end of the discussion period based on qualitative transcript analysis and/or ablations; see our response to Question 3 from Reviewer SMQo01.
> > >
> > > > if the effectiveness of EaE varies with the environment and the monitor (based on Fig 3), in what way should the readers view EaE as a useful contribution to the safety community? Since many existing methods also depend on these factors, how does EaE offer significant advantages beyond existing approaches?
> > >
> > > Practitioners don’t have to use a single monitoring approach in all situations. For example, follow-up work might use our results as a starting point to explore a router mechanism that uses different monitors in different contexts, knowing that EaE-like approaches are likely to be more useful in settings that involve coding tasks and reasoning models.
> > >
> > > Where EaE works well, the improvements are substantial. In some cases, we achieve more than a twofold improvement in catch rate at 1% FPR (Figure 4). Since coding tasks are among the most popular LLM applications, these gains are practically important. A routing system that directs coding transcripts to an EaE monitor while using other methods elsewhere would achieve strong performance improvements beyond existing approaches in some settings without trade-offs in others.
> > >
> > > Beyond context-dependent use, there are two further ways in which EaE might be a useful contribution to the community. First, our work can inform follow-up work in designing even better monitor architectures, demonstrating that hierarchical approaches to monitoring can be very effective in some settings. Second, EaE can be cheaply combined with stronger Evaluator models (Finding 4), potentially expanding the range of environments where it outperforms prior methods in a cost-controlled comparison beyond coding and MLE.
> > >
> > > > However, the authors draw this conclusion only from one setup (BigCodeBench-sabotage), and it is unclear to me whether this finding also generalizes to the other 3 setups in the paper. I would hope to see more evidence beyond one environment. And drawing a general conclusion from one environment seems overreaching.
> > >
> > > Due to compute limitations, it wasn’t possible for us to test reasoning model attackers in SHADE-Arena and MLE-sabotage. However, given that BigCodeBench-sabotage has a large number of tasks and samples, it seems reasonable to draw conclusions based on it alone: Arnav et al. performed all of their experiments in BigCodeBench-sabotage! Furthermore, the performance gap between CoT-only monitors on traditional and reasoning LLM trajectories is extremely significant: CoT-only catch rates drop from 65.6% for Claude-3.7-Sonnet:nonthinking to 29.7% for Claude-3.7-Sonnet:thinking and from 61.1% to 31.4% for DeepSeek-V3/R1 (Figure 5). We acknowledge that this conclusion is less robust than ones supported by experiments in all three environments, but the magnitude of the effect makes it a noteworthy result.
> > >
> > > We hope these answers address your concerns! Please let us know if you have any additional questions.

---

> > > > ### Comment · Reviewer_iJFr · 2025-11-20
> > > >
> > > > > EaE is also an information filtering technique, so when EaE performs better than the CoT+action monitor, that’s an instance of the less-is-more effect! (...)
> > > >
> > > > Thanks for the further explanation. I am now more convinced that there is a less-is-more effect given this. However, it still seems to me that there's no less-is-more effect in SHADE-Arena at all if you consider error bars. It would be great to see an explanation or some further analysis of why that's the case.
> > > >
> > > > > explore a router mechanism that uses different monitors in different contexts (...)
> > > >
> > > > Thank you for the clarifications, but I still do not find the argument sufficiently convincing. The core issue is that EaE’s effectiveness is highly context-dependent, with strong gains limited to certain environments and monitors. The suggestion that practitioners could rely on a router mechanism to determine when to apply EaE is not, in my view, a compelling justification, both because such routing is generally not a desirable or robust software engineering practice, and because it shifts the contribution to speculative future work rather than demonstrating present, broadly applicable value.
> > > >
> > > > > Due to compute limitations, it wasn’t possible for us to test reasoning model attackers in SHADE-Arena and MLE-sabotage. However, given that BigCodeBench-sabotage has a large number of tasks and samples, it seems reasonable to draw conclusions based on it alone (...)
> > > >
> > > > Thank you for the additional context, but I do not find the response fully convincing either here. While I understand the compute constraints, this does not resolve the central concern that the conclusion about CoT-only monitors is drawn from a single environment, making the generality of the finding unclear. Citing prior work that also used BigCodeBench does not justify extrapolating broader claims here, especially since the paper attempts to make a general claim about monitor behavior (Lines 92-95). Large effect sizes within one setup are great, but they do not substitute for cross-environment validation, particularly in safety evaluation, where results often vary across setups. As a result, I still view the conclusion as overreaching given the limited empirical evidence.
> > > >
> > > > I thank the authors for further clarifying the less-is-more effect and for their efforts to deepen our understanding of LLM monitors. However, given the concerns outlined above, even with the additional explanations provided in the rebuttal, I am only comfortable raising my rating to a 4.

---

> > > > > ### Author Response · Authors · 2025-12-03
> > > > >
> > > > > Thank you for raising the score and for the additional comments!
> > > > >
> > > > > > However, it still seems to me that there's no less-is-more effect in SHADE-Arena at all if you consider error bars.
> > > > >
> > > > > We acknowledge that information filtering techniques don’t outperform CoT+action monitoring in a statistically significant way in SHADE-Arena. However, CoT+action monitoring never outperforms information filtering in a statistically significant manner in SHADE-Arena either, meaning that no significant effect is observed either way. Thus, our experiments in SHADE-Arena don’t contradict our overall takeaway that information filtering techniques perform better, instead being a null result. We have updated the last paragraph of Section 1 to better reflect this.
> > > > >
> > > > > > The core issue is that EaE’s effectiveness is highly context-dependent, with strong gains limited to certain environments and monitors.
> > > > >
> > > > > Upon reflection, we agree that a router mechanism is not ideal for real-world deployments. However, context-specific monitors remain valuable in certain scenarios. For example, consider internal deployments where AI companies use models as internal research assistants instead of sharing them with the general public. In such settings, the majority of tasks are likely to involve coding and ML engineering—contexts where we have demonstrated EaE to be highly effective. Furthermore, even though the technique is somewhat environment-dependent, we observed strong gains across all monitor models (see Appendix B.1.1, Figure 9). Thus, we still believe that our results are valuable for the AI safety community.
> > > > >
> > > > > > Large effect sizes within one setup are great, but they do not substitute for cross-environment validation, particularly in safety evaluation, where results often vary across setups.
> > > > >
> > > > > We acknowledge the lack of cross-environment validation for this finding, and have updated the last paragraph of Section 1 to better reflect this.

---

### Official Review · Reviewer_2UvC · 2025-10-31

**Soundness:** 2
**Presentation:** 3
**Contribution:** 2
**Rating:** 4
**Confidence:** 3

**Summary:**

The paper studies the task of monitoring for sabotage: given an LLM agent prompted to perform a hidden task, can an LLM monitor detect the presence of this hidden task?

Following Arnav et al. (2025)’s finding that monitors with access to both CoT and actions can sometimes underperform action-only monitors, the paper proposes the “less-is-more” effect: sometimes, provisioning monitors with more information (e.g. CoT + actions) causes them to underperform monitors with less (e.g. just actions, or just CoT).

The paper experiments on four monitoring approaches: action-only, CoT-only, CoT+action, and their novel approach, extract-and-evaluate (EaE). EaA works by separating monitoring into two phases: the “Extractor” reasons about the transcript and extracts quotes, while the “Evaluator” produces a suspiciousness score based only on extracted inputs.

The paper runs experiments on three environments: SHADE-Arena, MLE-sabotage, and BigCodeBench-sabotage, with two monitor LLMs (Claude-3.7-Sonnet and GPT-4.1-mini), and seven attacker LLMs (two “traditional”/non-reasoning, five reasoning).

The paper demonstrates the less-is-more effect empirically, finding that CoT+action often underperforms other strategies, despite having the most information. It also finds that EaE sometimes outperforms other methods.

**Strengths:**

1. The abstract and introduction are clearly written, and the paper’s core claim about the less-is-more effect is easy to follow.
2. The proposed approach, extract-and-evaluate, is simple and well-motivated in light of the less-is-more finding.
3. Beyond the core claims about less-is-more and EaE, Section 5 gives a number of interesting empirical findings from the conducted experiments. I think these will be useful to researchers working on LLM monitoring.
4. The studied task, monitoring for sabotage, is an important problem for safety.

**Weaknesses:**

1. “extract-and-evaluate monitoring technique that improves sabotage detection rates in the BigCodeBenchsabotage environment by up to 12 percentage points over the next-best monitor
without increasing the false positive rate" - this feels like a misleadingly cherry-picked presentation of evidence in the abstract. EaE performs at a statistically indistinguishable level in many settings in Figure 3, and underperforms CoT only in BigBench traditional attacker/GPT-4.1. Without an a-priori reason to privilege this specific setting, you should present either aggregate statistics, or the full range from min to max.

2. I found it difficult to find the full details of any individual experiment. Which models are used as attackers, and which are used as monitors? Details appear spread throughout the Results section; Figure 4 shows five models, but others are listed in the appendix. It would be easier to follow if there was a concise list of all models used, and which roles they were used for, in the Methodology section.

3. I also found it difficult to follow the justification for some of the findings in section 5. I think it would be helpful to have a single graph for each finding, focusing on the features under study. For example, Finding 2 could show a scatterplot between monitor performance and average transcript length, with a point for each monitor-environment pair; Finding 7 could show a bar graph with monitor performance on reasoning models vs. monitor performance on traditional models. Figures could be in appendix, with links from each finding paragraph.

4. Limited novelty of the less-is-more effect itself, as prior work (Arnav et al. 2025) already empirically observed cases where CoT+action underperforms action alone.

5. While the less-is-more effect is demonstrated empirically, it’s still not clear to me *why* it happens. Is the problem due to distracting information? Or model performance in general declining over a certain context length? Given that prior work has already demonstrated this effect empirically, it would be a more valuable contribution to better understand its cause. E.g. if you add plausible-but-irrelevant text, does this impact performance more than adding the same number of clearly-irrelevant tokens (random characters, or just “...” repeated)?

**Questions:**

1. Figure 3, “error bars are bootstrapped 95% CIs” - how is the bootstrapping done? IIUC, Figure 3 aggregates over attacker models, but these attacker models may be evaluated on the same questions, so model-sample pairs aren’t IID, and a naive bootstrap will underestimate CIs. Could you provide additional details on how you’re computing CIs when aggregating over data? (See e.g. https://www.anthropic.com/research/statistical-approach-to-model-evals, “Cluster standard errors” for one statistical approach used in situations like this.)

2. Figure 6, “We use AUROC rather than log-AUROC for compactness.” - what does this mean?

3. For the results related to monitor awareness, the authors might be interested in the results of this paper, which specifically studies how monitor awareness impacts detection rates and behavior: https://openreview.net/forum?id=r4Q6o7KGdb

---

> ### Author Response · Authors · 2025-11-19
>
> Thank you for your thoughtful feedback!
>
> ---
>
> ## Weaknesses
> ### Cherry-picked evidence
> We agree that the abstract should have provided a more nuanced description of the results. In the new version of the paper, we have updated it to say “EaE rarely performs worse than other methods and achieves substantial improvements in some settings. For example, in BigCodeBench-sabotage with GPT-4.1-mini as the monitor model, EaE improves sabotage detection rates by 16.8 percentage points over the next-best monitor without increasing the false positive rate. In SHADE-Arena, EaE slightly outperforms the best baseline with a GPT-4.1-mini monitor and slightly trails with a Claude-3.7-Sonnet monitor.” Due to an error in how we weighted results by sample counts, we underestimated EaE's improvement in the original version; correcting this changes the figure from 12 to 16.8 percentage points.
>
> ### Confusing description of experiment details
> Thank you for pointing this out! We have updated the Methodology section with a paragraph giving a detailed overview of the used models.
>
> ### Lack of justification for findings
> Thank you for the recommendation to illustrate each finding with a separate plot! We will upload an updated version of the paper with those plots before the end of the review period.
>
> ### Limited novelty of the less-is-more effect
> While we agree that the less-is-more effect has been discussed in prior literature, we think that the scope and depth of our investigation substantially advances understanding of this phenomenon. While Arnav et al. (2025) observed the effect in a single setting (BigCodeBench-sabotage with reasoning model attackers) and for a single monitor (GPT-4.1-mini), we provide a systematic demonstration across three environments, both traditional and nonreasoning LLMs, four different monitor models, and different attacker awareness conditions.
>
> ### Lack of ablation studies
> We agree that we should do more to understand the causes of the less-is-more effect and are planning to perform qualitative transcript analysis to investigate them. If uncertainties remain after transcript analysis, we will also perform ablations that best resolve them.
>
> ---
>
> ## Questions
> ### Bootstrapping methodology
> Figure 3 actually does not aggregate over attacker models: in all environments, the attacker is Claude 3.7 Sonnet. Though we evaluated other attacker models in BigCodeBench-sabotage, the results for those models are displayed only in figures 4 and 5. We have emphasized this in the updated description of Figure 3.
>
> Our bootstrapping procedure resamples trajectory-monitor score pairs with replacement, using 1000 bootstrap samples. We don’t average over attackers and we perform the bootstrapping procedure separately for each monitor. However, we discovered that we overlooked a different issue that necessitates the use of cluster standard errors: in each environment, we sample multiple agent trajectories from the same main task – side task pair. We will recalculate the CIs and let you know when we have updated the paper with the corrections. Thank you for bringing this issue to our attention!
>
> ### AUROC vs. log-AUROC
> We use AUROC in Figure 6 to reduce the scale of the y-axis, thus making the plot easier to follow. Since we generally use log-AUROC as our main metric throughout the paper, we found it important to clarify this. We rephrased the caption to say, “We use AUROC over log-AUROC because the takeaways are similar and the plot is more readable.”
>
> ### Monitor awareness
> Thank you for mentioning this paper! We agree that this paper is relevant to our work and have added a citation to it. We also think that our results on awareness retain their relevance: in contrast to Li et al., we study a different phenomenon (malicious side task vs. sandbagging) and a different setting (long agentic environments vs. multiple-choice questions), and it is thus important to show that a dynamic similar to the one observed by Li et al. applies in our environments.
>
> ---
>
> We hope that these answers address your concerns! If you're satisfied with our responses, we'd be grateful if you'd consider updating your score.

---

> ### Comment · Reviewer_2UvC · 2025-11-24
>
> Thanks for your responses.
>
> **W1.** Thanks, I think this is a more accurate description of the results.
>
> **W2.** Thanks, I think this improves readability.
>
> **W3.** Pending the updated version.
>
> **W4.** I agree that the scope and depth of this investigation goes beyond Arnav et al. 2025. But given that the contribution is primarily an empirical one, rather than conceptual or methodological, I think careful justification of findings and analysis of potential causes of the observed impacts are particularly important.
>
> **W5.** Pending transcript analysis/ablations.
>
> **Q1.** Pending updates.
>
> **Q2.** Thanks, this is clearer.

---

> > ### Author Response · Authors · 2025-12-03
> >
> > Thank you for the response! We have now addressed your remaining concerns:
> > - **W3:** Instead of creating an appendix, we have improved the clarity of the Results section sufficiently that this appendix now seems unnecessary to us. An overview of our changes is given in our most recent top-level comment.
> > - **W5:** We have added Appendix E that provides a summary of the qualitative transcript analysis we performed. This has substantially reduced our uncertainties about the causes of the less-is-more effect. We provide an overview of the takeaways in our most recent top-level comment.
> > - **Q1:** We have adopted cluster standard errors from Miller (2024) as suggested. Our approach is explained in the description of Figure 3. An exception is made for SHADE-Arena, where we encountered some issues with the implementation of this approach. We will fix this for the camera-ready version.

---

### Official Review · Reviewer_uPer · 2025-10-31

**Soundness:** 3
**Presentation:** 3
**Contribution:** 2
**Rating:** 6
**Confidence:** 4

**Summary:**

This paper considers the problem of monitoring LLM agents for planning and executing malign side tasks alongside benign main tasks. The main goal of the paper is to evaluate which information the monitor should utilize and how that information should be utilized in order to maximize catch rates (while minimizing false positives) of the monitors. The authors introduce a novel two-step monitoring protocol, extract-and-evaluate (EaE), to improve the accuracy of weaker monitors. The authors evaluate various monitoring protocols against both traditional LLMs and reasoning models on three evaluation environments. The authors main finding is that more information is not always better, and developing various filtering methods for monitoring can be extremely beneficial.

**Strengths:**

* The main contribution of the paper is valuable to the AI control community: "naively applying monitors on the full agent trajectory is often insufficient to achieve optimal monitor performance."
* EaE seems to provide a Pareto improvement (performance / cost) over other monitoring strategies.
* The paper is very well-written and the presentation is overall well-done.

**Weaknesses:**

* Do not run against adaptive agents (i.e. where the agent can optimize itself knowing that it will be monitored). This makes the evaluation setup unrealistic in comparison to actual deployment scenarios. Figure 3 shows that when the agent knows more about the monitoring environment (e.g. SHADE-Arena) the monitoring performance decreases dramatically, and the proposed approaches provide much less benefit (especially when using a weaker monitor). This indicates that the proposed approaches will not necessarily be useful in practical AI control scenarios.
* Some of the results are quite obvious. For example, Finding 2 and Finding 6 are both very intuitive given the "lost-in-the-middle" effect [1], especially for weaker models. I.e. it makes sense that less-is-more especially when monitoring with a weaker agent. Nevertheless, even given this intuition, it is important to show the result empirically.
* Some of the claims are not obviously backed up by the data. For example, the authors claim that EaE monitoring improves catch rates by 12 percentage points over the next best catch rates (line 321, and mentioned earlier in the paper). Can the authors point to where this can be seen. It is not obvious from the figures in the main paper.
* There's no one monitor setup that always wins, and the increase in performance is not always significant.


[1] Liu, N. F., Lin, K., Hewitt, J., Paranjape, A., Bevilacqua, M., Petroni, F., & Liang, P. (2023). Lost in the middle: How language models use long contexts. arXiv preprint arXiv:2307.03172.

**Questions:**

* How is agent success determined? Is it a deterministic function based on the evaluation environment?
* Why do CoT-only (w/o actions?) monitors perform better on traditional LLMs (non-reasoning models)?
* What is the monitor in Figure 4? Is it GPT-4.1-mini?
* Can you provide a more in-depth justification for why filtering for success makes sense? Why would the CoT-only monitor have a large advantage? This is not obvious.
* Furthermore, if the actions made by the agent could cause harm, would it not make sense to catch the agent before harm was caused? I think this might be related to the online vs offline setting (Roger and Shlegeris (2025)).
* What are task success rates for the various models and evaluation environments?
* Can you further justify why the experimental setup is a pragmatic proxy for the real-world AI control scenario? In practice, agents will not necessarily be prompted to perform these side actions and therefore might also tend to hide their true intentions more (because misalignment).

---

> ### Author Response · Authors · 2025-11-19
>
> Thank you for your insightful feedback!
>
> ---
>
> ## Weaknesses
> ### Adaptive agents
> Our agents are adaptive to a degree: in all environments, we studied a variation where the agent is told in a prompt that it will be monitored (see Sec. 5.4 and App. B.1.3). One could go further and allow agents to optimize themselves against specific monitors, but this is not standard practice in AI control literature. Since our work was focused on improving blue-team info-filtering strategies, studying such adaptive agents was out of the scope of our paper.
>
> ### Lost-in-the-middle effect
> We agree that the lost-in-the-middle effect provides a partial explanation for our results and have added a reference to the paper. However, that effect was discovered back in 2023 for models significantly weaker yet than GPT-4.1-mini, so showing that long-context degradation is still a significant issue for monitors is an important result. Furthermore, knowledge of the lost-in-the-middle effect alone is insufficient to infer the quantitative impact of the effect on monitor performance.
>
> ### 12 percentage points claim
> Thanks for pointing out this issue! We have updated Fig. 4 to provide a full comparison to Arnav et al. (2025) in the main body of the paper and to explicitly display the improvement in the catch rate for each attacker model.
>
> ### No single monitor wins
> We agree that the results depend on the environment and monitor. However, this applies to many methods used in practice. We think that a systematic study of the strengths and weaknesses of different methods is still valuable: for example, follow-up work might use our results as a starting point to explore a router mechanism that uses different monitors in different contexts.
>
> ---
>
> ## Questions
> ### 1
> In MLE-sabotage and BigCodeBench-sabotage, agent success is a deterministic function of the environment. E.g., MLE-sabotage filters out incorrectly formatted trajectories, as well as ones where the agent encountered time-outs or token limits in the scaffold. SHADE-Arena uses a mix of task-specific deterministic functions and LLM graders.
>
> ### 2
> Our current understanding is that CoT-only monitors perform better on traditional LLMs due to the transcripts of traditional LLMs being dramatically shorter (see App. D3: traditional LLM transcripts have lengths of ~400-800 tokens in BigCodeBench-sabotage, while ones produced by reasoning LLMs have ~3,500-8,000 tokens). When a traditional LLM reasons about a side task, it’s easy for the monitor to notice that and there’s little room for the attacker to produce explanations that confuse or mislead the monitor.
>
> ### 3
> Yes, it is GPT-4.1-mini. We have updated the figure description of Figure 4 to clarify this.
>
> ### 4
> We filter for success to ensure that the agent makes a strong attempt at both its main and side tasks. Not filtering for side task success would allow for cases where the agent does not attempt the side task at all and there is nothing for the monitor to catch, even though the trajectory counts as a malicious one where the presence of a side task would be expected. It would also allow for cases where the agent discusses the side task in its CoT but is unable to find a solution, thus not performing actions that lead to side task completion and giving an advantage to monitors with CoT access. The claim that this advantage would only apply to the CoT-only monitor was incorrect—it applies to all kinds of CoT monitors. We have fixed this mistake in the updated version of the paper.
>
> ### 5
> We agree that in some situations, it’s important to catch the agent before its actions can possibly cause harm. However, the offline setting is also relevant: many safety failures can arise in low-stakes situations where no single action is concerning on its own but the combined impact of many actions lead to catastrophic outcomes [1]. As we expect many of the insights from the offline setting to transfer to online monitoring and as the offline setting involves more baselines from past literature, we decided to make that our focus, but have mentioned online monitoring as an important follow-up direction in Sec. 6.
>
> ### 6
> Filtering for both main and side task completion, the success rates are between 25-40% in all three environments.
>
> ### 7
> The settings we use are the SOTA control evals, covering a range of realistic settings including coding, MLE, and personal assistant tasks. We utilize prompts from existing work which have been optimized to elicit sabotage from agents. We agree that this is only a proxy for how real-life misaligned agents would act, but creating new evaluations was not the focus of our work. The settings we chose are the best proxy currently available, as prompting agents to perform side tasks is standard practice in the AI control literature.
>
> We hope our reply addresses your concerns and merits an improved score!
>
> [1] Christiano, P. (2021). Low-stakes alignment. https://alignmentforum.org/posts/TPan9sQFuPP6jgEJo/low-stakes-alignment

---

> > ### Comment · Reviewer_uPer · 2025-11-25
> >
> > Thank you for your response. The rebuttal has not changed my opinion of this work. I will maintain my score.

---

### Official Review · Reviewer_SMQo · 2025-11-01

**Soundness:** 2
**Presentation:** 2
**Contribution:** 2
**Rating:** 4
**Confidence:** 4

**Summary:**

This paper studies how what the monitor is allowed to see (actions, CoT, both, or a filtered subset) changes an LLM monitor’s ability to catch covert sabotage in three “AI control” environments—SHADE‑Arena (assistant tasks with hidden side‑goals), BigCodeBench‑sabotage (single‑step coding with backdoors/side rules), and MLE‑sabotage (agentic ML engineering). The central empirical claim is a “less‑is‑more” effect: monitors given less raw information sometimes detect sabotage better than those given both CoT and actions. Motivated by this, the authors introduce Extract‑and‑Evaluate (EaE): a two‑stage protocol where an Extractor first quotes salient snippets from (CoT + actions), and then an Evaluator scores suspiciousness using only those quotes. EaE improves catch rates in BigCodeBench‑sabotage by up to 12 percentage points on average (and 18.2 pp on subtle side tasks) at 1% FPR, and can amplify weaker monitor models cost‑effectively. The paper further shows that CoT‑only monitors often outperform action‑only ones (contrary to some earlier results), that agents who believe their CoT is private are far easier to catch, and that on a cost–performance frontier, Gemini and Claude families with information filtering dominate. Methodologically, the paper emphasizes log‑AUROC (AUROC with the FPR axis in log‑space starting at 1%) to reflect low‑FPR operational regimes.

**Strengths:**

A clear empirical story around information access. The paper asks a practical question: "what should your monitor actually look at?" and answers it with cross‑environment evidence that naively giving monitors everything (CoT+actions) is often suboptimal, with CoT‑only or extracted subsets performing better. This is useful guidance for safety teams building pipeline monitors today.

---
A simple, actionable protocol (EaE) with measurable gains. EaE operationalizes the less‑is‑more intuition via hierarchical filtering: have one model extract quotes before a second model scores only those quotes. In BigCodeBench‑sabotage, EaE beats competing monitors on TPR@1% FPR (overall +12 pp; +18.2 pp on subtle tasks), and a “weak Extractor + strong Evaluator” variant achieves large lifts at modest additional cost. This is a pragmatic contribution that teams can test quickly.

---
Agent awareness and monitorability. The finding that telling the agent its CoT is private dramatically improves CoT‑based monitoring (Δ log‑AUROC > 0.4 in SHADE‑Arena) is an important reminder that situational awareness policies matter. It also partially reconciles mixed results in the literature: unaware agents verbalize more, so CoT monitors work better.

---
Cost–performance perspective. The Pareto analysis is helpful: Gemini and Claude occupy much of the efficient frontier, and information filtering appears on many Pareto‑optimal points; gains diminish after roughly $0.005/transcript, but spending ~$0.1 ensures robust performance giving practical ballparks for deployment.

---
Care with evaluation metrics. Focusing on log‑AUROC (weighted to ≤10% FPR) reflects real auditing budgets; the authors also reproduce results under standard AUROC and discuss bootstrap variance and threshold effects.

**Weaknesses:**

**Contribution framing**: Beyond EaE, other “contributions” (awareness, cost analysis, reasoning vs. traditional LLMs) feel like observations rather than core advances.

**CoT-only novelty claim**: The authors state that their CoT-only results are novel because prior work (e.g., Arnav et al., 2025) focused on action-only vs. CoT+action monitors. This seems fair, but the claim could be clearer: CoT monitoring itself is well-known, so the novelty lies in the systematic evaluation across environments, not in introducing CoT-only monitoring per se. Claiming novelty here is misleading.

**Scope of EaE**: Gains hinge on very long CoTs overwhelming weak monitors. If CoTs are truncated or monitors scale better, EaE’s advantage largely disappears. If CoT length were capped or chunked, would EaE still matter? The authors acknowledge that on non-subtle tasks and traditional LLMs, EaE is not better than prior methods.

**Mechanistic gap**: Less-is-more may reflect parsing limits and post-hoc justification bias, but no controlled ablations isolate these factors.

**Questions:**

1. Adversarial extracts. How robust is EaE if the Extractor misses the key quotes or is adversarially distracted by benign snippets? Any recall metric for the Extractor (fraction of ground‑truth incriminating spans captured)?

2. When total token cost is fixed, does EaE still outperform CoT+action or hybrid? Also, what happens if you swap roles (strong Extractor, weak Evaluator)?

3. Mechanistic clarity: Did you run ablations to confirm that failures of CoT+action monitors are due to post-hoc justifications rather than other confounders (e.g., prompt complexity)? Can you disentangle causes of less-is-more? For example:

* Truncate CoTs to fixed token budgets.
* Remove post-hoc justification phrases.
* Shuffle or mask irrelevant CoT spans.
* How do these affect monitor rankings?


4. Evaluator necessity: Why is a separate Evaluator needed? Did you test a single-step prompt that extracts quotes and assigns a suspicion score? If not, why assume two stages outperform one?


5. Scope of novelty: CoT-only monitoring is presented as a novel contribution, but prior work has studied it. Can you clarify what’s new here?

6.  If CoT length were limited to what the monitor can handle (or processed in chunks), would EaE still outperform? Is the gain mostly due to mitigating long-context degradation?

---

> ### Author Response · Authors · 2025-11-19
>
> Thank you for the thoughtful feedback!
>
> ---
>
> ## Weaknesses
> ### Contributions
> Observations can be contributions! It seems quite standard in ML to contribute primarily empirical observations that can support multiple hypotheses. For example, [1] focuses on the observed phenomenon of emergent misalignment.
>
> ### CoT-only novelty claim
> We agree that the claim of novelty may be misleading. Since systematic CoT-only evaluation isn’t a central contribution, we have removed the claim.
>
> ### Scope of EaE
> We disagree that gains hinge on very long CoTs overwhelming monitors. We observe the largest gains in BigCodeBench-sabotage with reasoning models, where transcripts are ~3,500-8,000 tokens long (see Fig. 4 and App. D3). The gains in SHADE-Arena are much smaller, despite an average transcript length of ~20,000 tokens.
>
> We acknowledge that EaE is less effective for very short transcripts, such as those of traditional LLMs in BigCodeBench. We have added a new figure in the updated paper (Fig. 5) to further stress this. However, these transcripts are ~500-800 tokens long, which is unrepresentative of real-world transcripts. Frontier LLMs have context windows of up to 1M tokens, making it infeasible to cap monitor inputs to 400-800 tokens. The wider adoption of reasoning models further increases the load of monitors. This makes our results on reasoning models particularly important.
>
> For non-subtle side tasks, EaE performs similarly to the best of the four baselines across all tested models (see Fig. 4). Though it doesn’t outperform prior methods by a large margin, it’s still the most consistent method.
>
> ### Mechanistic gap
> See answer to question 3.
>
> ---
>
> ## Questions
> ### 1
> We studied adversarial policies in BigCodeBench-sabotage and SHADE-Arena by giving the attacker hints that help it circumvent the monitor. Though those attack policies were not optimized to specifically target EaE, they provided strategies for making the snippets discussing the side task more benign, e.g. by framing the side task as a necessary intermediate step for completing the main task. Since such strategies did not lead to a reduction in catch rates (see App. C4 and D4), we didn’t pursue them further.
>
> We don’t think the lack of adversarial extracts is an important weakness: attackers are instructed to be sneaky when pursuing a side task by default and most trajectories contain long benign snippets unrelated to the side task. The fact that EaE often outperforms other methods on such transcripts implies that getting distracted by benign snippets is less of a concern with EaE compared to prior methods.
>
> ### 2
> > When the total token cost is fixed, does EaE still outperform other monitors?
>
> We show in Fig. 7 that when accounting for token costs in MLE-sabotage, EaE appears on the Pareto frontier more often than any other monitor type.
>
> > What happens if you use a strong Extractor and weak Evaluator?
>
> Thank you for suggesting this experiment! We ran it and present the results in App. D5 of the updated paper. We found that using GPT-4.1 as a strong Extractor and GPT-4.1-mini as a weak Evaluator has comparable performance to using GPT-4.1-mini as both, implying that the Evaluator’s capabilities are more important than the Extractor’s.
>
> ### 3
> Thank you for these suggestions! We agree we should do more to disentangle causes of less-is-more. We think the first step is to do qualitative transcript analysis, then to perform ablations to study remaining uncertainties. There are many possible ablations, including those suggested by you and reviewer 2UvC: we likely cannot do all of them, but please let us know if any particular ablation stands out to you as especially important.
>
> ### 4
> The CoT+action monitor in BigCodeBench-sabotage is functionally equivalent to a single-step EaE monitor: it first reasons about the transcript, then extracts quotes, and then outputs the score. This monitor performs substantially worse than the EaE monitor, implying that separating extraction and evaluation into two stages is necessary.
>
> ### 5
> See answer to weakness 2.
>
> ### 6
> In BigCodeBench with traditional LLMs, the setting with the shortest transcripts, EaE still outperforms CoT+action monitors (see Fig. 5). Limiting real-world deployments to even shorter transcripts is infeasible (see our answer to weakness 3). Evaluating monitors that process the CoT in chunks was outside the scope of this project (see the top of page 5), but we encourage future work to study this.
>
> Finally, we would like to note that other reviewers have commented very positively on the presentation of our paper. E.g., reviewer uPer31 wrote: “The paper is very well-written and the presentation is overall well-done.” Could you provide some pointers for what you would like to see improved about the paper’s presentation?
>
> We hope our answers address your concerns and justify a higher score!
>
> [1] Betley et al., 2025. Emergent Misalignment: Narrow finetuning can produce broadly misaligned LLMs. arXiv:2502.17424

---

> > ### Comment · Reviewer_SMQo · 2025-11-25
> >
> > Thank you for your answers. I appreciate the clarifications and the additional experiment on Extractor/Evaluator roles. However, my main concerns remain:
> > * The paper still lacks a mechanistic explanation for why two stages help. The rebuttal states that CoT+action is “functionally equivalent” to a single-step EaE and performs worse, but this only shows an empirical difference and it does not clarify why the separation matters.
> > * Several claimed contributions (awareness effect, lost-in-the-middle) feel incremental. Observations can be contributions, but their value depends on novelty and generality. Here, many observations replicate known phenomena in slightly different settings rather than offering new conceptual insights.
> >
> > Given these unresolved issues, I will maintain my original rating as the contribution remains primarily incremental and lacks deeper analysis of mechanisms.

---

> > > ### Author Response · Authors · 2025-12-03
> > >
> > > Thank you for outlining your further concerns!
> > >
> > > > The paper still lacks a mechanistic explanation for why two stages help.
> > >
> > > We have addressed this concern in the updated version of the paper. We have added Appendix E with a qualitative transcript analysis. Our main takeaway is that the less-is-more effect is largely driven by it being easier for the monitor to refer to the agent’s instructions and check its reasoning for deviations from those instructions when the length of the reasoning trace is shorter. Thus, an Extractor that filters out most of the agent transcript makes it easier for the Evaluator to check the quotes for deviations from the instructions. This isn’t possible when the extraction and evaluation steps are combined in a single context window, as the Evaluator must have both the agent’s and Extractor’s full reasoning traces in its context in that case.
> > >
> > > > Several claimed contributions (awareness effect, lost-in-the-middle) feel incremental. Observations can be contributions, but their value depends on novelty and generality. Here, many observations replicate known phenomena in slightly different settings rather than offering new conceptual insights.
> > >
> > > While we acknowledge that some of our findings have been noted in other contexts in prior literature, we still disagree that they don’t offer any new conceptual insights. Even though LLMs were known to struggle with long contexts beforehand, the quantitative impact of this on monitor performance was not known: for example, it was quite surprising to us that monitor performance improved twofold in BigCodeBench-sabotage when the transcript length was cut from ~7000 to <1000 tokens, either by using the EaE monitor or by replacing a reasoning LLM attacker with a traditional LLM attacker. Additionally, as we mentioned in our response to reviewer 2UvC, the scope of our analysis of the less-is-more effect is substantially wider than that of previous work, spanning three environments, both traditional and nonreasoning LLMs, four different monitor models, and different attacker awareness conditions.

---

### Author Response · Authors · 2025-11-19
**Overview of updates to the paper**

Dear reviewers,

We appreciate all of the thoughtful feedback we’ve received! We have updated the paper accordingly and uploaded an improved version. Below, we’ll give an overview of the main changes:

1. **Clearer figures.** We have split Figure 4 from the old version of the paper into two separate figures, Figure 4 and Figure 5, to improve the readability of the figures and to ensure that no figure corresponds to more than two findings. The paper now displays one figure for each of the five results sections. Additionally, we have created improved versions of each of the figures: for example, Figure 4 now explicitly shows the catch rate delta between EaE and the previous SOTA from [1], while Figure 7 is larger and significantly more readable. Finally, the descriptions of Figure 3 and Figure 4 are now more detailed and unambiguous.
2. **Updated abstract.** Based on feedback from reviewer 2UvC, we have updated the abstract of the paper to provide a more nuanced overview of the performance of EaE.
3. **Fixed improvement margins in Section 5.2.** Due to an error in how we weighted results by sample counts, we underestimated the relative improvements over baselines from the use of EaE in BigCodeBench-sabotage. We have updated the abstract and Section 5.2 to present the correct figures. Section 5.2 now also provides a more detailed description of the calculation.
4. **New EaE experiment.** We have added a subsection to Appendix D that compares the performance of an EaE monitor with a strong Extractor and weak Evaluator to the performance of an EaE monitor with a weak Extractor and strong Evaluator, as suggested by reviewer SMQo01. We found that the capability of the Evaluator model impacts the EaE monitor’s performance more than the capability of the Extractor model.
5. **Improved Methodology section.** We have added two paragraphs to the Methodology section that provide a detailed overview of the models used in our experiments.
6. **Removal of claims with insufficient justification.** We have improved the abstract and introduction to paint a more holistic picture of the strengths and weaknesses of EaE.

We are planning to make further improvements before the end of the review period, with two priorities:
1. **Transcript analysis and ablations.** We are planning to perform qualitative transcript analysis to better understand the causes of the less-is-more effect. If we have remaining uncertainties about the causes of the effect after this analysis, we will run additional ablations, such as those suggested by reviewers SMQo01 and 2UvC, to narrow down the mechanism.
2. **An appendix displaying one plot for every finding.** As suggested by reviewer 2UvC, we will add an appendix that illustrates each finding with a separate plot.

We hope that these improvements increase the quality of the paper. If they address your concerns, we'd be grateful if you'd consider updating your score.

[1] Arnav et al., 2025. CoT Red-Handed: Stress Testing Chain-of-Thought Monitoring. arXiv:2505.23575

---

### Author Response · Authors · 2025-12-03
**Overview of additional updates to the paper**

Dear AC,

We have uploaded a new version of the paper that further addresses the reviewers’ concerns. Below is an overview of the main changes:

1. **Qualitative transcript analysis.** We have added Appendix E that provides a summary of the qualitative transcript analysis we performed. This has substantially reduced our uncertainties about the causes of the less-is-more effect: we now believe that the less-is-more effect is largely driven by it being easier for the monitor to refer to the agent’s instructions and check its reasoning for deviations from those instructions when the length of the reasoning trace is shorter. This also explains the difference in the CoT-only monitor’s performance between reasoning and traditional LLMs (Section 5.3): when evaluating the transcript of a traditional LLM, the reasoning trace that the CoT-only monitor needs to check for deviations from the instructions is up to ten times shorter. In cases where EaE fails, the failure can be caused both by the side task being insufficiently salient to the Extractor and by miscalibrated judgements by the Evaluator, though the latter is more common unless the Evaluator model is more capable than the Extractor (Appendix D.5).

2. **Fixed confidence intervals.** Based on reviewer 2UvC’s suggestion, we have adopted cluster standard errors and updated all figures with error bars calculated with this methodology. Our approach is explained in the description of Figure 3. An exception is made for SHADE-Arena, where we encountered some issues with the implementation of this approach. We will fix this for the camera-ready version.

3. **Clearer structure for Section 5.** Reviewer 2UvC suggested adding an appendix with a single plot for each finding. Instead of this appendix, we restructured the Results section to eliminate this need. Each subsection of Section 5 now contains exactly two findings and one corresponding figure that illustrates both findings. We have also improved the clarity of the figures to ensure that all of them reflect both of their corresponding findings. These changes have substantially improved the clarity of the Results section and, in our opinion, made the suggested appendix unnecessary.

We believe that these changes, combined with those outlined in our previous top-level comment, have substantially improved the quality of the paper compared to the initial submission. We hope that you find these revisions have sufficiently strengthened the paper.

---

### Meta-Review · Area_Chair_xmno · 2026-01-02

**Summary:**

The reviewers have raised the following major concerns:
1. The work's contributions and novelty need reframing. (SMQo, uPer, 2UvC, iJFr)
2. The method's effectiveness hinges on long CoTs. (SMQo)
3. Further mechanistic studies are required. (SMQo, 2UvC)
4. Lack of evaluation on adaptive agents. (uPer)
5. Some findings are trivial. (uPer)
6. The detection improvement may be overclaimed. (uPer, 2UvC)
7. Detailed experiment descriptions are needed. (2UvC)

**Reviewer Concerns:**

Concerns addressed by rebuttal:
(2) (7)

Outstanding concerns:
(1) (3) (4) (5) (6)

**Reviewer Scores:**

- SMQo: The rebuttal partially addresses the reviewer's comments; They would likely keep their score.
- uPer: The rebuttal partially addresses the reviewer's comments; They would likely keep their score.
- 2UvC: The rebuttal partially addresses the reviewer's comments; They would likely keep their score.
- iJFr: The rebuttal partially addresses the reviewer's comments; They would likely keep their score.

---

### Decision · Program_Chairs · 2026-01-26

Reject